# Spherical Silver Nanoparticles Located on Reduced Graphene Oxide Nanocomposites as Sensitive Electrochemical Sensors for Detection of L-Cysteine

**DOI:** 10.3390/s24061789

**Published:** 2024-03-10

**Authors:** Fei Hua, Tiancheng Yao, Youzhi Yao

**Affiliations:** 1School of Materials Engineering, Wuhu Institute of Technology, Wuhu 241003, China; huafei@whit.edu.cn; 2College of Ecology and Environment, Anhui Normal University, Wuhu 241003, China; 3College of Chemistry and Chemical Engineering, Anhui University, Hefei 230601, China; c41914016@stu.ahu.edu.cn

**Keywords:** L-cysteine, electrochemical sensor, graphene, nanocomposite, detection

## Abstract

A new, simple, and effective one-step reduction method was applied to prepare a nanocomposite with spherical polycrystalline silver nanoparticles attached to the surface of reduced graphene oxide (Ag@rGO) at room temperature. Equipment such as X-ray diffraction (XRD), scanning electron microscopy (SEM), transmission electron microscopy (TEM), X-ray photoelectron spectroscopy (XPS), and Fourier transform infrared spectroscopy (FTIR) was used to characterize the morphology and composition of the Ag@rGO nanocomposite. A novel electrochemical sensor for detecting L-cysteine was proposed based on fixing Ag@rGO onto a glassy carbon electrode. The electrocatalytic behavior of the sensor was studied via cyclic voltammetry and amperometry. The results indicate that due to the synergistic effect of graphene with a large surface area, abundant active sites, and silver nanoparticles with good conductivity and high catalytic activity, Ag@rGO nanocomposites exhibit significant electrocatalytic activity toward L-cysteine. Under optimal conditions, the constructed Ag@rGO electrochemical sensor has a wide detection range of 0.1–470 μM for L-cysteine, low detection limit of 0.057 μM, and high sensitivity of 215.36 nA M^−1^ cm^−2^. In addition, the modified electrode exhibits good anti-interference, reproducibility, and stability.

## 1. Introduction

L-cysteine, also known as L-2-amino-3-mercaptopropionic acid with the molecular formula C_3_H_7_NO_2_S, is a common amino acid in living organisms and an important amino acid and glycogenic amino acid required by the human body [1]. L-cysteine, as a low-molecular-weight amino thiol, can bind with some functional enzymes to form an active site called cysteine protease, which participates in many life processes and is an important substance for intracellular signal transduction and gene regulation [2,3]. L-cysteine is used for detoxification and regulation of biological oxidative balance, and it is an important biomarker, which can serve as a diagnostic indicator for various diseases. As a therapeutic agent and antioxidant, L-cysteine is widely used in industries such as medicine, food, and cosmetics [4]. For example, L-cysteine is an amino acid detoxifying agent, which participates in the reduction process of cells and phospholipid metabolism in the liver. It has a pharmacological effect of protecting liver cells from damage, promoting liver function recovery, and strengthening the liver [5]. Abnormal L-cysteine indicators indicate a potential risk of disease in the human body. For example, low levels of L-cysteine can cause liver damage, muscle and fat loss, heart disease, hair color loss, and other diseases; high levels of L-cysteine can lead to increased risks of renal failure, diabetes, Alzheimer’s disease, and Parkinson’s disease [6,7].

The L-cysteine measurement technology for clinical and commercial purposes is still under research, and the development of low-cost, simple, and highly sensitive L-cysteine detectors has become a key task in the food, biological, and medical fields [8]. At present, there are many detection methods for L-cysteine, mainly including high-performance liquid chromatography [9], colorimetric detection [10], flow injection analysis (FIA) [11], photoelectrochemical (PEC) sensors [12], fluorescence spectroscopy [13], and electrochemical detection techniques [14,15,16]. Among these methods, most of them have drawbacks, such as a complex sample preparation process, expensive precision instruments, inconvenient operation, or low sensitivity, which greatly limit their application in practical detection. Electrochemical methods have attracted much attention due to their simple operation, low cost, portability, and high sensitivity.

The application of traditional electrodes in practical detection is limited due to defects such as easy surface passivation, high oxidation overpotential required for detecting L-cysteine, and low sensitivity of voltammetry response [17,18,19]. One solution to address these defects is to use various electrochemical media and nanomaterials to modify the electrode surface, increase the electrode surface area, accelerate the electron migration rate, and enhance the electrochemical response [20,21,22,23].

Due to its superior properties, such as a large specific surface area, high electron mobility, and excellent mechanical resistance, research on graphene and its derivatives has continued to increase since its discovery. Extensive study has been conducted on the attachment of metals or metal oxides onto graphene, such as platinum (Pt), gold (Au), silver (Ag), copper (Cu), titanium oxide (TiO_2_), zinc oxide (ZnO), manganese dioxide (MnO_2_), nickel oxide (NiO), etc. It is expected that nanocomposites will be strengthened or improved in the required properties [24,25,26,27,28,29,30,31]. Compared with Pt and Au, Ag is very economical and has excellent conductivity. It can reduce the electrode resistivity and improve its electron transfer rate, making it an excellent candidate for electrode materials. Silver nanoparticles embedded between graphene sheets can prevent the re-accumulation of graphene layers during the reduction process, increasing the electroactive surface area of graphene nanocomposites [32]. The synthesis and application research of Ag-GO nanocomposites has attracted much attention [33,34,35,36]. Jeong et al. prepared a nanocomposite of Ag-doped rGO and studied its electrochemical performance. The results showed that adding Ag to rGO increased the specific capacitance value by about 25 times [33].

Nanocomposites—such as metal nanoparticles, graphene quantum dots, MoS_2_ quantum dots—have become promising materials for sensing and imaging biogenic sulfur in living cells [37]. L-cysteine is a sulfur-containing amino acid with high affinity to the Ag (111) surface [38], and its thiol side chains are adsorbed on noble metal surfaces at room temperature [39]. Here, we proposed a one-pot reduction method for preparing Ag attached on rGO using sodium borohydride (NaBH_4_) as a reducing agent at room temperature. Compared to others, the chemical reduction method has been widely used due to its simple operation, high efficiency, and ability to obtain various types of precursors [40]. The prepared nanocomposite was characterized by morphology and structure and was used to modify the glassy carbon electrode (GCE) for electrochemical detection of L-cysteine. The results showed that the prepared sensor exhibited good electrochemical response to L-cysteine, with good stability and sensitivity, low detection limit, wide linear range, and excellent reproducibility and stability.

## 2. Experimental Section

### 2.1. Reagents and Equipment

L-cysteine, AgNO_3_, NaBH_4_, polyethylene glycol (PEG-400), ethanol, nitric acid, and nafion, all of analytical grade, were purchased from Aladdin Chemical Reagents (Shanghai, China) and used without any further purification. Graphene oxide was prepared using an improved Hummers’ method. Phosphate buffer solutions with various pH values (0.10 M) were prepared using Na_2_HPO_4_ and NaH_2_PO_4_. All solutions were prepared with double-distilled water (18.25 MΩ cm resistivity).

A 524G digital constant temperature magnetic stirrer, CT14D centrifuge, Fourier transform infrared spectrometer (FT-IR850), and BeNano 180 Zeta potential analyzer were purchased from Shanghai Bangwo Instrument Equipment Co., Ltd., Shanghai, China. The surface morphology and elemental content distribution spectrum of the nanocomposite were studied using scanning electron microscopy (SEM) (Hitachi, s-4800, Tokyo, Japan) and energy-dispersive spectroscopy (EDS) (GG314-JPS-9200). The microstructure of the nanocomposite was characterized using transmission electron microscopy, and the lattice structure and spacing of carbon and silver on the sample surface were detected using high-resolution transmission electron microscopy (TEM, JEOL, JEM 2100F).

The electrochemical experiment was conducted using a CHI660E electrochemical analyzer (Chenhua Instrument Co., Ltd., Shanghai, China) and a traditional three-electrode battery. The working electrodes were, respectively, glass carbon electrodes (GCE) (Φ = 3 mm), rGO-modified electrode (rGO/GCE), and Ag@rGO/GCE. The reference electrode was an Ag/AgCl (saturated KCl) electrode, with a platinum electrode as the auxiliary electrode. Before each experiment, the solution was blown with purified nitrogen gas for 15 min to remove oxygen. Unless otherwise specified, all measurements were carried out at room temperature.

### 2.2. One-Step Synthesis of Silver @ Reduced Graphene Oxide Nanocomposite

Water-soluble graphene oxide (3 mg/mL) was fabricated following an improved Hummers’ method using commercial graphene (99.9%). In a typical experiment, 5 mL of polyethylene glycol 400 was added as a surfactant to 50 mL of GO solution to prevent aggregation of silver nanoparticles.

AgNO_3_ (4 mL, 0.15 M) was slowly dropped into the mixture above and stirred magnetically for 30 min, with the aim of fully adsorbing Ag^+^ by GO and coordinating more Ag^+^ with the active sites (carbonyl, hydroxyl, carboxyl) on the surface of GO. Under continuous stirring, newly prepared NaBH_4_ (0.06 M) was mixed dropwise into the solution, with a dosage of approximately 5 mL. The black substance was deposited at the bottom of the container after 6 h of stewing. After the supernatant was removed, the black sediment was subsequently washed with ethanol and deionized water, until the filtrate became neutral, and then dried in high vacuum at 50 ℃ for 2 h.

### 2.3. Preparation of Modified Electrodes

Before modification with the nanocomposite, the surface of the glassy carbon electrode was polished with chamois and 0.05 μm alumina slurry. The processed electrodes were sonicated in 1:1 nitric acid for 5 min to remove any remaining alumina slurry from the surface, washed with distilled water, and then dried at room temperature. In order to fix the nanocomposite on the surface of the glassy carbon electrode, after 1 mg of Ag@rGO was dispersed into 2 mL of Nafion solution, the mixture was sonicated for 5 min to obtain the Ag@rGO/Nafion dispersion solution. Subsequently, 10 μL of the dispersed solution was coated onto the surface of GCE and dried in atmosphere to obtain Ag@rGO-modified glassy carbon electrode (Ag@rGO/GCE). GCE modified with rGO (rGO/GCE) was prepared via a similar process.

## 3. Results and Discussion

### 3.1. Characterization

#### 3.1.1. Morphology and Component Characterization

The SEM technique was used to characterize the morphology of Ag@rGO nanocomposite. As shown in Figure 1a, a large number of grape-shaped (spherical) silver nanoparticles were distributed on the wrinkled surface of rGO. It can be seen in Figure 1b that the phase composition of rGO and Ag@rGO nanocomposite was verified using XRD. The X-ray diffraction pattern of rGO shows a sharp (002) peak centered at 26.3°. The characteristic peaks (2θ) of Ag@rGO are situated at 38.1°, 44.3°, 64.4°, 77.5°, and 81.5°, indexed to the (111), (200), (220), (311), and (222) crystal planes of metallic Ag, respectively. According to JCPDS 00-004-0783, silver nanoparticles have a face-centered cubic structure, consistent with previous reported literature [41,42,43]. The average particle size of Ag NPs is 20.3 nm, according to the Debye–Scherrer formula D = 0.9 λ/(β cos θ) [44].

To further confirm the composition elements of the composite material, EDX was used to characterize the Ag@rGO nanocomposite. The EDX image shown in Figure 1c contains strong peaks of silver and carbon as the main constituent element of the composite, except for the silicon peak from the sample supporting substrate silicon wafer, while the oxygen peak displayed is weak, indicating adequate reduction in silver ions and graphene oxide.

Figure 2a depicts the TEM micrograph of rGO, with obvious wrinkles on its surface but no other substances adhering to it. Figure 2b shows the TEM image of Ag@rGO nanocomposite, and it is evident that spherical silver nanoparticles were dispersed on rGO sheets, with a size range of 10–40 nm. Parts of the silver nanoparticles are stacked together, consistent with the SEM results. The high-resolution TEM (HRTEM) images of Ag@rGO in Figure 2c clearly demonstrate the presence of Ag in the (111) (d111 = 0.2378 nm) and (200) plane (d200 = 0.2064 nm), respectively. The selected area electron diffraction (SAED) pattern is shown in Figure 2d, which is composed of four concentric rings with different radii. The formation of this circular pattern may be due to the presence of a large number of disorderly oriented particles. Small crystal particles in the sample area under electron beam irradiation indicate that the silver nanoparticles belong to a polycrystalline structure [45].

The chemical states of Ag and carbon in Ag@rGO were determined under XPS. Figure 3a shows the high-resolution C1s spectrum of GO, with three peaks located at 284.6 eV (C=C), 286.7 eV (C-OH), and 288.4 eV (O=C-OH), denoting the presence of different functional groups (double bonds, hydroxyl groups, and carbonyl groups) on the GO sheet [46]. Based on the blue curve (Ag@rGO) in the insert image in Figure 3a, it can clearly be seen that the intensity of the C=C peak does not change significantly, while the peaks of C-OH and O=C-OH almost disappear. As displayed in Figure 3b, compared to the O1s of GO, the peak value of O1s (Ag@rGO) significantly decreased, indicating that the vast majority of oxygen-containing functional groups in GO were reduced. Figure 3c shows the peaks related to C1s, Ag 3d, and O1s in the Ag@rGO nanocomposite. Compared to the XPS spectrum of GO, the C1s and Ag 3d peaks dominate in Ag@rGO, and the peak intensity of O1s is greatly reduced. Figure 3d shows the high-resolution XPS of the 3d peak of Ag. The peaks mainly occur at 368.5 eV (3d5/2) and 374.6 eV (3d3/2), with spin energy separation values of approximately 6 eV between the two peaks confirming that silver is in a zero-valence state on the surface of rGO sheets [47].

Figure 4 shows the FTIR spectra of GO and Ag@rGO. It is evident that after the reduction reaction, the characteristic peaks of C-O (1121 cm^−1^) and C=O (1796 cm^−1^) in GO almost disappear in the FTIR of Ag@rGO. All results show that GO and AgNO_3_ were successfully reduced in one step at room temperature to the Ag@rGO nanocomposite.

#### 3.1.2. Characterization of Modified Electrodes

Electrochemical impedance spectroscopy (EIS) is a convenient and effective tool for studying electrode interface properties. The ideal impedance spectrum consists of a semicircle at high frequencies and a straight line at low frequencies. The semicircle diameter of the EIS spectrum is related to the electron transfer resistance (Rct), revealing the electron transfer kinetics of the redox probe at the electrode interface, and the straight line is a typical feature of diffusion process control [48]. Figure 5 illustrates the Nyquist plots of the electrochemical properties of bare GCE (curve a), rGO/GCE (curve b), and Ag@rGO/GCE (curve c) under the presence of 5.0 μM K_3_Fe(CN)_6_/K_4_Fe(CN)_6_ and 0.1 M KCl. Obviously, rGO-modified GCE exhibits a smaller interfacial electron transfer resistance than bare GCE, which is due to the increased electron transfer rate on the electrode surface by rGO. Compared with bare GCE and rGO/GCE, the impedance of the Ag@rGO/GCE nanocomposite is further reduced, which may be attributed to the excellent conductivity of Ag nanoparticles.

Fit the obtained impedance data using a Randles circuit (illustrated in Figure 5), including the electron transfer process (charge transfer resistance Rct) and diffusion process (Warburg-type impedance) [49]. The electron transfer resistance (Rct) is estimated to be 495 Ω, 378 Ω, and 206 Ω, corresponding to GCE, rGO/GCE, and Ag@rGO/GCE, respectively. The results indicate that Ag@rGO/GCE possesses good interfacial electron transfer performance, which is beneficial for constructing highly sensitive electrochemical sensors.

### 3.2. Electrochemical Response of L-Cysteine on Different Modified Electrodes

The electrochemical detection performance of L-cysteine in phosphate-buffered solution (pH = 7.0) was emulated via cyclic voltammetry (CV) using GCE, rGO/GCE, and Ag@rGO/GCE. Figure 6 displays the cyclic voltammetry of different electrodes in response to L-cysteine (50 μM). Obviously, GCE has no electrocatalytic activity toward L-cysteine, since no significant redox peak is observed. In the potential range of −0.40–0.60V, a strong oxidation peak appears at +0.0316 V, and a very weak reduction peak appears at −0.3247 V on the CV diagram of rGO/GCE, indicating that the redox process of L-cysteine on the electrode is almost irreversible. In the CV pattern of Ag@rGO/GCE, a strong oxidation peak appears at −0.0208 V; a weak reduction peak appears at −0.3452 V; and the oxidation overpotential decreases. Usually, the electrochemical determination of L-cysteine requires a high oxidation overpotential. For example, Wang et al. [50] reported that when using Pt/Fe_3_O_4_ nanoparticles/reduced-graphene-oxide-modified GCE to detect L-cysteine, an irreversible CV peak at +0.65V was observed. Cao et al. [51] reported that an electrochemical oxidation potential of +0.522V was observed on an electrode modified with gold nanoparticles. The above experimental results indicate that Ag@rGO/GCE-modified GCE has excellent electrocatalytic performance for L-cysteine. As shown in the inserted figure, none of the three electrodes showed significant redox peaks for the phosphate-buffered solution (pH = 7.0) without L-cysteine sample.

### 3.3. Effect of Scanning Rate on Oxidation Reduction in L-Cysteine

Figure 7 shows the CV curves of L-cysteine (50 μM) on Ag@rGO/GCE at different scanning rates. It can be seen that in the range of 20–180 mV s^−1^, the peak current of Ag@rGO/GCE in the oxidation process of L-cysteine rises significantly with the increase in scanning rate. When the scanning rate exceeds 120 mV s^−1^, the value of the peak current increases slightly. If the relationship between peak current and scanning speed satisfies the Randles–Ševčík equation, Ip = (2.69 × 10^5^)n^3/2^ADCʋ^1/2^ (n, number of electrons transferred during the redox process; A, electrode area, generally a geometric area; D, diffusion coefficient; C, ion concentration participating in the reaction; V, scanning rate set by cyclic voltammetry), i.e., a square root relationship, this illustrates that the electrocatalytic process is a diffusion-controlled reaction. The illustration in Figure 7 shows the relationship between the peak redox current and the square root of the scanning rate, which is linear in the range of 20–180 (mV s^−1^) scanning rate, with a correlation coefficient R of 0.98282, further indicating that the electrode reaction process is controlled by the diffusion of L-cysteine, which is consistent with the electrochemical oxidation mechanism of L-cysteine on solid electrodes proposed by Ralph et al. [52]. In addition, as the scanning rate increases, it can be observed that the weak reduction current peak of L-cysteine gradually increases. It may be concluded that L-cysteine oxidation products cannot completely desorb from rGO with the scanning rate increasing, leading to a reduction process.

### 3.4. Effect of pH on L-Cysteine Electrocatalysis

In order to optimize the experimental conditions for L-cysteine detection, the effect of Ag@rGO/GCE on the electrochemical behavior of L-cysteine at pH values from 5.0 to 9.0 was investigated using the CV method. As shown in Figure 8, the oxidation peak current increased with the increase in pH (5.0–7.0); meanwhile, when the pH value exceeded 7.0, the peak current value decreased. This indicates that under the condition of pH = 7.0, Ag@rGO/GCE exhibits the best electrocatalytic performance for the oxidation of L-cysteine. Therefore, in order to obtain the optimal current response conditions, experiments exploring the electrochemical behavior of Ag@rGO/GCE for L-cysteine were conducted in a phosphate-buffered solution with pH = 7.0.

### 3.5. Amperometric Detection of L-Cysteine at Ag@rGO/GCE

Under steady-state conditions, the current method is a simple and sensitive electrochemical detection method [4]. In an optimized state, applying a potential of −0.0208 V and continuous stirring to promote solution homogenization, the typical amperometric response signals obtained by continuously adding different concentrations of L-cysteine are shown in Figure 9. At the initial stage, low concentrations of L-cysteine were added, and a steady-state current response was obtained within 2–5 s, implying a fast electron transfer process in the electrode. Subsequently, in the following steps, samples of the same concentration were added three times every 50 s, with concentrations of 0.1, 0.5, 1, 5, 10, 20, 40, and 80 μM, respectively. L-cysteine exhibits a significant linear increase in current response. The sensor possesses a wide linear range from 0.1 μM to 470 μM (S/N = 3), and the linear regression equation is as follows: *i*_Pa_ (μA) = 0.33058 + 0.16115 ***C***_L-cysteine_ (μM). As shown in the inserted figure, the R is equal to 0.99449; sensitivity is 215.36 nA M^−1^ cm^−2^; and the detection limit (LOD) is 0.057 μM. Table 1 lists the detection parameters of some other electrochemical sensors for L-cysteine. Compared to them, the detection range and limit of Ag@rGO/GCE for L-cysteine were significantly improved. The results indicate that the Ag@rGO nanocomposite can be used as an excellent electrode material for detecting L-cysteine, which can be attributed to the large surface area, rich active sites, high conductivity of silver, and excellent catalytic activity of rGO. In addition, in a 40 mL phosphate-buffered solution (pH = 7.0), the Zeta potential of Ag GO nanocomposite was measured to be −16.5 mV, indicating that Ag-supported graphene nanomaterials have excellent stability [53], promoting the collision and binding of L-cysteine with quantum dots on the material [38] and undergoing rapid oxidation reactions under the catalysis of Ag nanoparticles.

### 3.6. Influence of Interference, Reproducibility, and Stability

To demonstrate the selectivity of the Ag@rGO/GCE sensor, the potential interferences of some typical organic compounds for the detection of L-cysteine were also investigated using the amperometric method. As can be seen in Figure 10, when 50 μM L-cysteine was added to the test sample, there was a significant change in the current response. When 200 μM interferents—such as glucose, L-tyrosine, citric acid, tryptophan, sodium oxalate, and alanine—were added, the current response value hardly changed. This indicates that Ag@rGO/GCE has good anti-interference ability in detecting L-cysteine.

The reproducibility and stability test of the sensor is shown in Figure 11. In a phosphate-buffered solution (pH = 7.0), the relative standard deviation (RSD) of 50 μM L-cysteine detected using the amperometric method for seven consecutive measurements was 3.7% (less than 5%). The sensor was stored at room temperature and in a dry atmosphere, and 50 μM L-cysteine was detected every 2 days. After 20 days, the current response value of the sensor was observed, which was 9.2% lower than the initial value, indicating the good long-term stability of Ag@rGO/GCE.

As an example of practical application, AgrGO/GCE was performed to detect L-cysteine in the presence of human serum samples. Serum was diluted 10 times in a phosphate-buffered solution (0.1 M, pH = 7.0) as a test sample, and a certain amount of L-cysteine standard solution (0, 5, 10, and 20 μM) was added to the human serum sample through the standard addition method. The results are shown in Table 2, and they are consistent with the reported concentration range of cysteine in serum (30–200 M) [58]. The corresponding recovery rate results range between 98.5% and 103.3%, and the relative standard deviation (RSD) ranges between 1.8% and 4.7%. This indicates that Ag@rGO/GCE has a satisfactory practical application for L-cysteine detection in real samples.

## 4. Conclusions

In this work, we successfully synthesized the Ag@rGO nanocomposite and constructed an electrochemical non-enzymatic sensor for L-cysteine detection. The electrochemical research results indicate that the sensor exhibits excellent electrocatalytic oxidation ability toward L-cysteine in a 0.1 M phosphate-buffered solution at pH = 7.0. Under optimized conditions, Ag@rGO/GCE exhibits excellent electrocatalytic activity as a L-cysteine sensor, with a wide linear range, low detection limit, high sensitivity, and fast response time, which can be attributed to the high surface area, abundant active sites, and accelerated electron transfer rate of the Ag@rGO nanocomposite. In addition, Ag@rGO/GCE has the advantages of simple preparation and high stability; therefore, it has broad application prospects for detecting L-cysteine in actual samples.

## Figures and Tables

**Figure 1 sensors-24-01789-f001:**
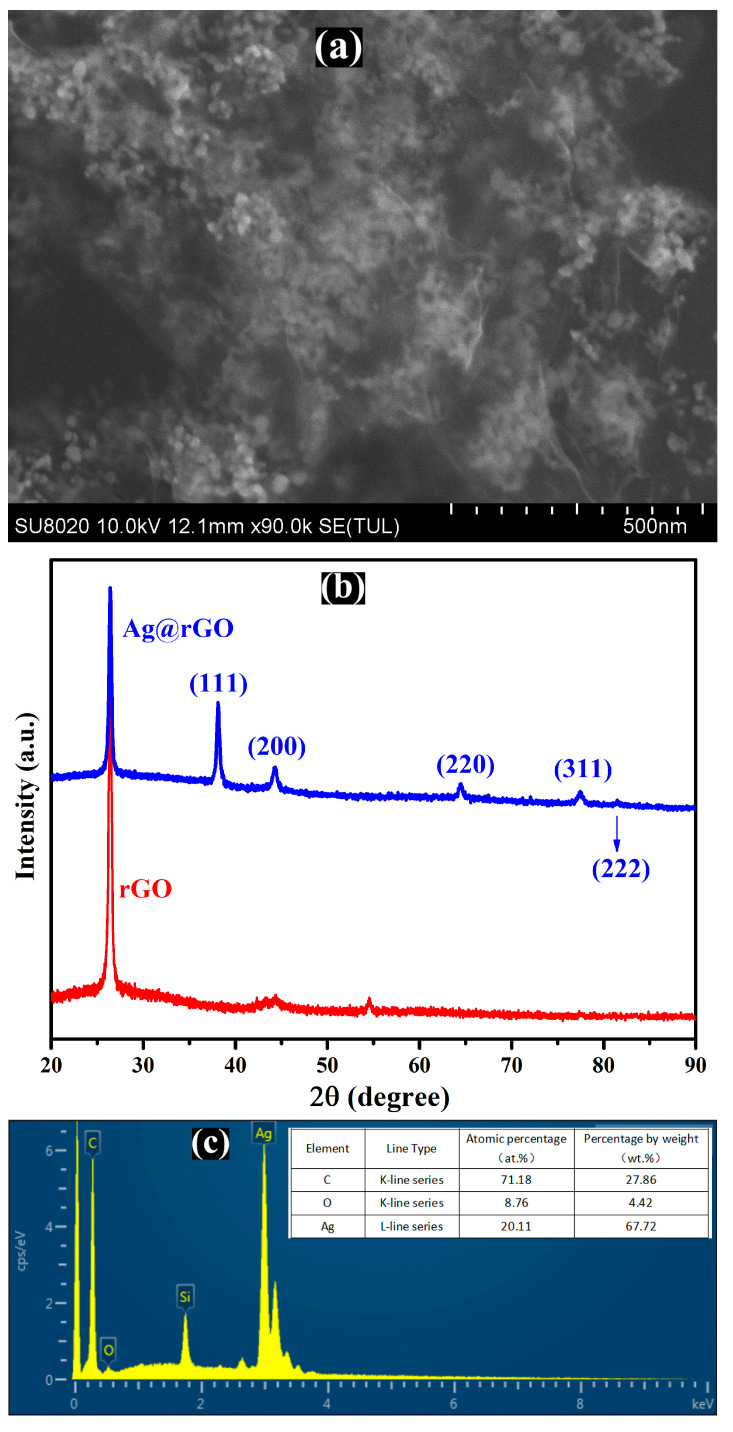
SEM image (**a**), XRD pattern (**b**), and EDX spectrum (**c**) of Ag@rGO.

**Figure 2 sensors-24-01789-f002:**
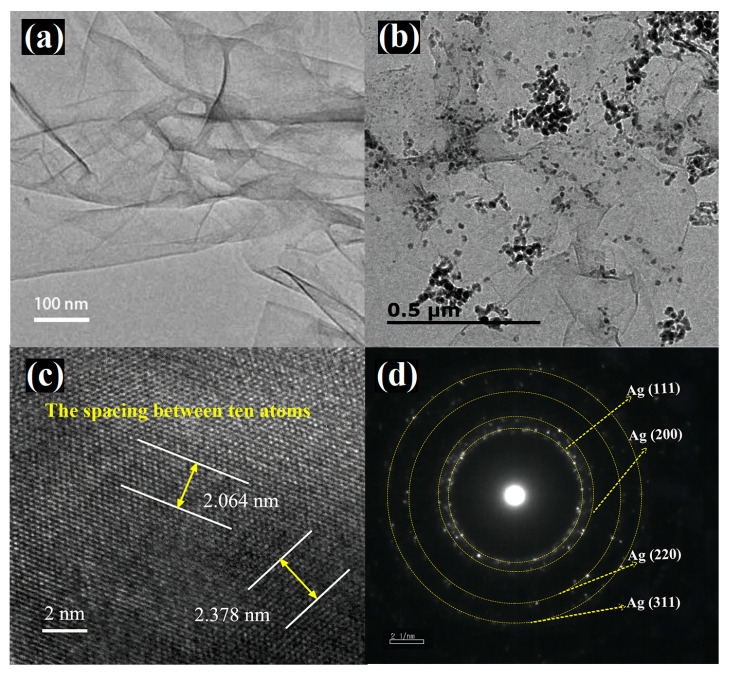
TEM images of GO (**a**) and Ag@rGO (**b**); HRTEM image (**c**) and SAED pattern (**d**) of Ag@rGO.

**Figure 3 sensors-24-01789-f003:**
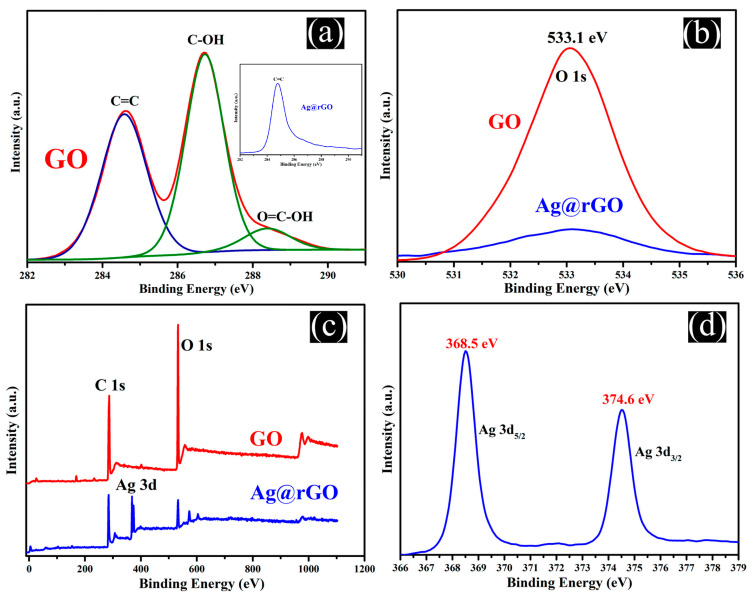
XPS spectra of rGO and Ag@rGO: (**a**) High-resolution deconvoluted scan spectra of C 1s, (**b**) Scan spectra of O 1s, (**c**) Survey scan and (**d**) High-resolution scan spectra of Ag 3d.

**Figure 4 sensors-24-01789-f004:**
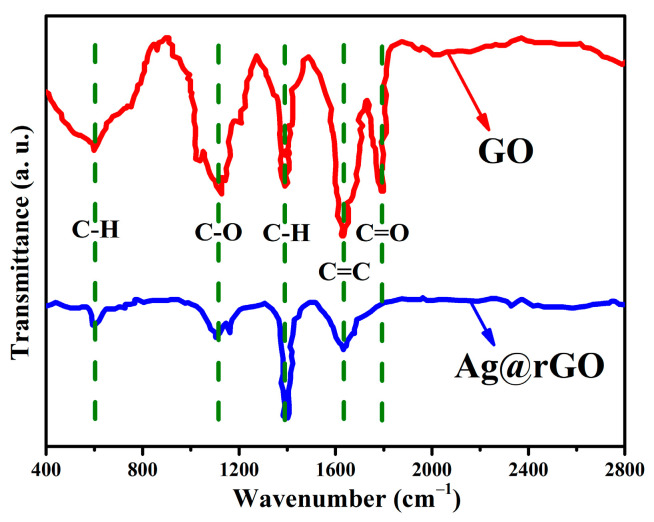
FTIR spectra of GO (red curve) and Ag@rGO nanocomposite (blue curve).

**Figure 5 sensors-24-01789-f005:**
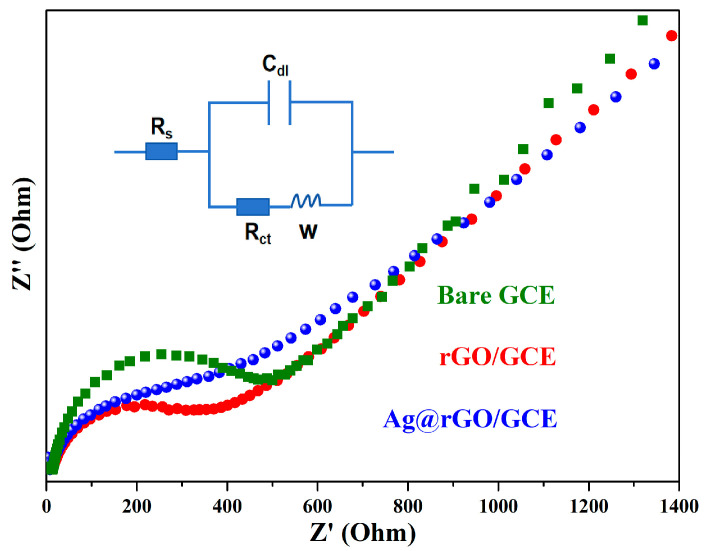
Electrochemical impedance spectroscopy (EIS) of bare GCE, rGO/GCE, Ag@rGO/GCE under 5.0 μM K_3_Fe(CN)_6_/K_4_Fe(CN)_6_ (1:1).

**Figure 6 sensors-24-01789-f006:**
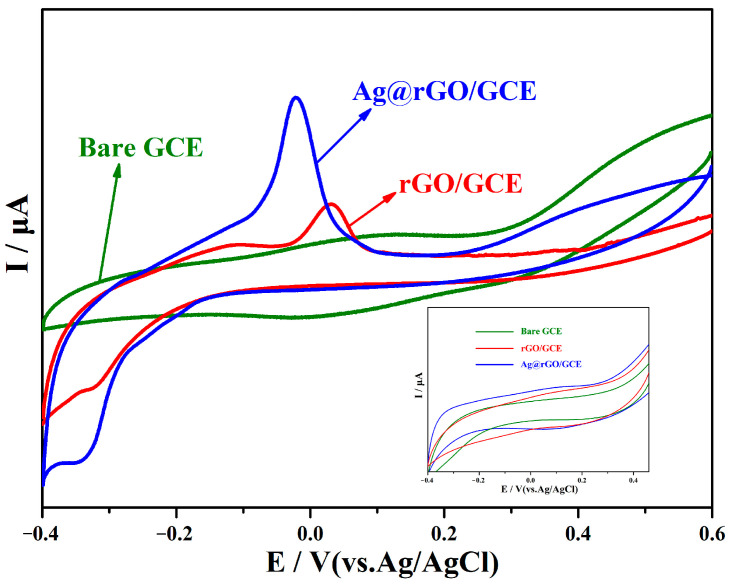
CV curves of 50 μM L-cysteine on bare GCE, rGO/GCE, and Ag@rGO/GCE in 0.1 M phosphate-buffered solution (pH = 7.0). Scanning rate: 100 mV s^−1^.

**Figure 7 sensors-24-01789-f007:**
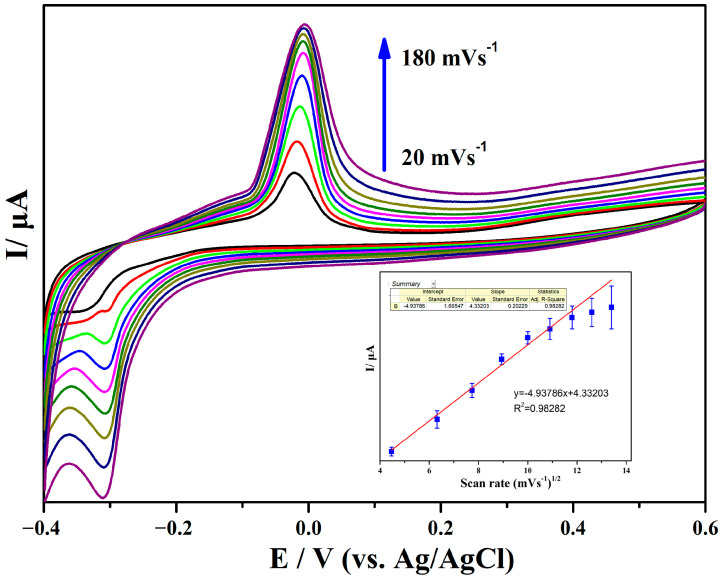
Effect of scanning rate on the voltammograms of Ag@rGO/GCE in pH = 7.0 phosphate-buffered solution at 20, 40, 60, 80, 100, 120, 140, 160, and 180 mV s^−1^. Inset: Plot of the peak currents (I) vs. square root of the scanning rate.

**Figure 8 sensors-24-01789-f008:**
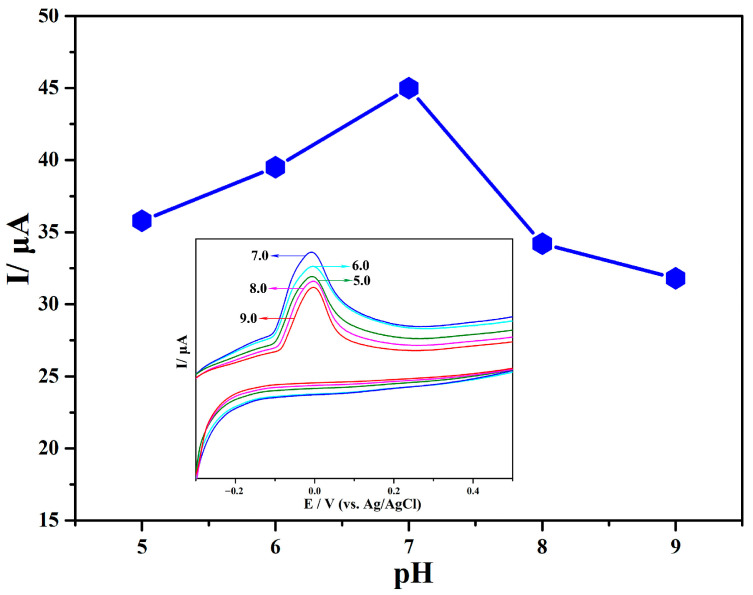
Cyclic voltammograms of electro-oxidation of 50 μM L-cysteine in 0.1 M PBS at different pH: 5.0, 6.0, 7.0, 8.0, 9.0. Scanning rate: 100 mV s^−1^.

**Figure 9 sensors-24-01789-f009:**
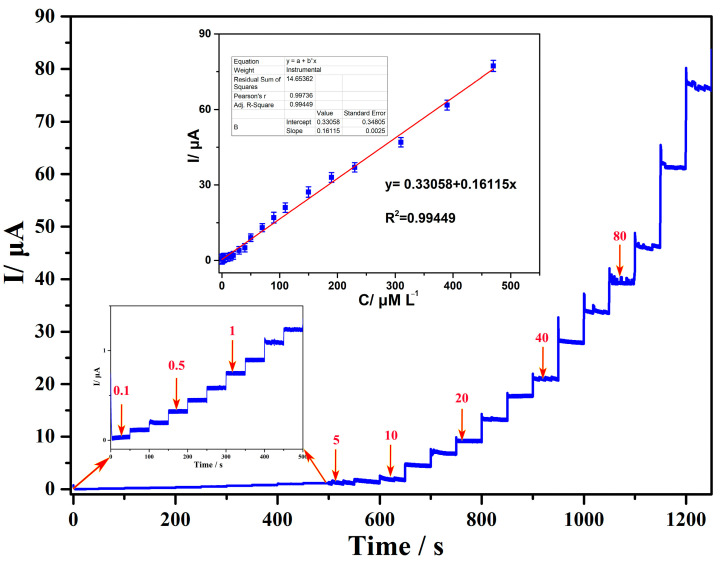
Amperometric responses at Ag@rGO/GCE with successive additions of L-cysteine in 0.1 M phosphate-buffered solution (pH = 7.0). Inset: Plots of enlarged curves showing the addition of 0.1–1 μM of L-cysteine and the corresponding calibration curve.

**Figure 10 sensors-24-01789-f010:**
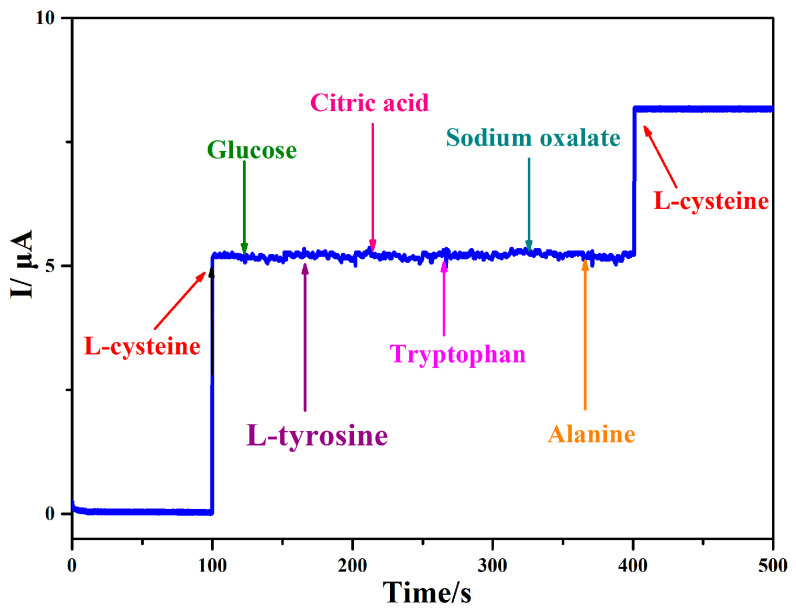
Amperometric response observed following additions of 50 μM L-cysteine and 200 μM various interfering substances.

**Figure 11 sensors-24-01789-f011:**
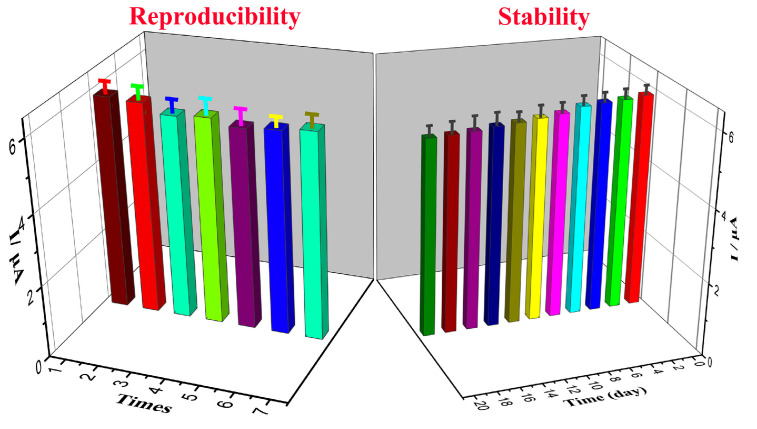
Reproducibility and stability of Ag@rGO/GCE for L-cysteine determination; the error bars illustrate the standard deviations of three independent measurements.

**Table 1 sensors-24-01789-t001:** Comparison for L-cysteine determination at Ag@rGO/GCE with reported electrodes.

Electrodes	Linear Range (μM)	Detection Limit (μM)	Ref.
CuFe_2_O_4_/reduced graphene oxide/Au nanoparticles modified GCE	50–200	0.383	[54]
Pt/Fe_3_O_4_ nanoparticles/reduced grapheme oxide modified GCE	100–1000	10	[50]
BiPr composite nanosheets modified GCE	1–2000	0.21	[15]
Pt/carbon nanotubes modified electrode	0.5–100	0.3	[18]
Au/Nafion modified GCE	2–80	1.0	[55]
Multi-walled carbon nanotubes modified GCE	10–500	5.4	[56]
Boron-doped carbon nanotubes modified GCE	0.78–200	0.26	[57]
This work	0.1–470	0.057	

**Table 2 sensors-24-01789-t002:** L-cysteine detection in real samples (n = 3).

Sample	Add (μM)	Found (μM)	Recovery (%)	RSD (%)
Serum	0	5.17 ± 0.42	–	4.7
	5	10.26 ± 0.36	103.3	3.1
	10	15.78 ± 0.65	98.5	2.6
	20	25.42 ± 0.31	99.2	1.8

## Data Availability

Data are contained within the article.

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
