# Peer review of "Spherical Silver Nanoparticles Located on Reduced Graphene Oxide Nanocomposites as Sensitive Electrochemical Sensors for Detection of L-Cysteine"

_sensors, 2024, doi:10.3390/s24061789_

Round 1

Reviewer 1 Report

Comments and Suggestions for Authors

The paper“Spherical silver nanoparticles located on reduced graphene oxide nanocomposites as sensitive electrochemical sensor for detection of L-cysteine ” is about a method of synthesizing silver @ reduced graphene oxide and applied to the determination of L-cysteine. In my opinion, there are some places needing to be revised:

First, The innovation of this study is not enough. Only from the article, I did not realize the advantages of this study compared with other methods for detecting L-cysteine.

Second, The method for measuring the electrochemical signal requires further explanation. In this study, cyclic voltammetry was used for condition optimization, while chronocurrent method was used for linear output and specificity verification. It should be explained whether the results obtained by these two methods in this study are consistent and reproducible.

Last, please embellish all over the paper.

Comments on the Quality of English Language

The quality of English in this article is generally as expected, with relatively concise descriptions and no obvious grammatical errors.

Reviewer 2 Report

Comments and Suggestions for Authors

Although the study titled "Spherical silver nanoparticles located on reduced graphene oxide nanocomposites as sensitive electrochemical sensor for detection of L-cysteine" seems suitable for publication in general terms, it is envisaged that the following arrangements be made.

1. Although the Ag@rGO synthesis method is well explained, the SEM images in Figure 1 do not reflect the morphology of the structure well. For this, it is expected to add only SEM images of graphene oxide and Ag-coated graphene oxide structure and indicate the orientation of Ag particles on the graphene surface.

2. XRD analysis showing the reduced state of the initially used graphene oxide and the graphene oxide combined with Ag is expected to be shown.

3. There is a resolution problem in SEM and XRD figures.

4. It is convenient to demonstrate in an FTIR analysis that the Ag particle binds to the carbonyl, hydroxyl, or carboxyl groups of graphene oxide.

Reviewer 3 Report

Comments and Suggestions for Authors

Comments to the author

Author has prepared Spherical silver nanoparticles located on reduced graphene oxide nanocomposites as sensitive electrochemical sensor for detection of L-cysteine, this study was well carried over, the figures were well structurally organized and arranged.

I have few concerns and comments that need to be clarified and justified prior to publication.

Round 2

Reviewer 2 Report

Comments and Suggestions for Authors

It is seen that the author revised his work by taking into consideration the issues I have previously mentioned. Therefore, there is no harm in publishing the study in your journal.

Reviewer 3 Report

Comments and Suggestions for Authors

Thank you for your explanations and answer's. I accept the work for further progress.